# Classification of Planetary Nebulae through Deep Transfer Learning

**Dayang N. F. Awang Iskandar** [1,2,*,†], **Albert A. Zijlstra** [2,†], **Iain McDonald** [2,3],
**Rosni Abdullah** [4], **Gary A. Fuller** [2], **Ahmad H. Fauzi** [1] and **Johari Abdullah** [1]

1   Faculty of Computer Science and Information Technology, Universiti Malaysia Sarawak,
    Sarawak 94300, Malaysia; hadinata@unimas.my (A.H.F.); ajohari@unimas.my (J.A.)
2   Jodrell Bank Centre for Astrophysics, Department of Physics and Astronomy, School of Natural Sciences,
    University of Manchester, Oxford Road, Manchester M13 9PL, UK; albert.zijlstra@manchester.ac.uk (A.A.Z.);
    Iain.Mcdonald-2@manchester.ac.uk (I.M.); gary.a.fuller@manchester.ac.uk (G.A.F.)
3   School of Physical Sciences, The Open University, Walton Hall, Kents Hill, Milton Keynes MK7 6AA, UK
4   School of Computer Sciences, Universiti Sains Malaysia, Pulau Pinang 11800, Malaysia; rosni@usm.my
*   Correspondence: dnfaiz@unimas.my
†   These authors contributed equally to this work.

**Abstract:** This study investigate the effectiveness of using Deep Learning (DL) for the classification of planetary nebulae (PNe). It focusses on distinguishing PNe from other types of objects, as well as their morphological classification. We adopted the deep transfer learning approach using three ImageNet pre-trained algorithms. This study was conducted using images from the Hong Kong/Australian Astronomical Observatory/Strasbourg Observatory H-alpha Planetary Nebula research platform database (HASH DB) and the Panoramic Survey Telescope and Rapid Response System (Pan-STARRS). We found that the algorithm has high success in distinguishing True PNe from other types of objects even without any parameter tuning. The Matthews correlation coefficient is 0.9. Our analysis shows that DenseNet201 is the most effective DL algorithm. For the morphological classification, we found for three classes, Bipolar, Elliptical and Round, half of objects are correctly classified. Further improvement may require more data and/or training. We discuss the trade-offs and potential avenues for future work and conclude that deep transfer learning can be utilized to classify wide-field astronomical images.

**Keywords:** deep learning; transfer learning; planetary nebulae; morphology; classification; HASH DB; Pan-STARRS

---

## 1. Introduction

A planetary nebulae (PN) forms when a sun-like star ejects its envelope at the end of its life. The ejected envelope forms an expanding nebula around the remnant core of the star which ionizes it. After some $10^4$ years, the PN fades from view, both because of the expansion and dilution of the nebula and because of the fading of the ionizing star. Around 3000 PNe are known in the Galaxy. PNe show up as compact nebulosity on images of the sky, with typical spectra that are dominated by emission lines. They are commonly identified by comparing images taken at different wavelengths. However, they can be confused with other types of astronomical objects: confirmation that a nebula is indeed a PN requires follow-up spectroscopy. A significant fraction of cataloged PNe were later found to be misidentified. An overview of PNe discovery surveys can be found in Parker [1].

The most up-to-date catalog of PNe in our Milky Way Galaxy is the Hong Kong/Australian Astronomical Observatory/Strasbourg Observatory H-alpha Planetary Nebula research platform

database (HASH DB) [2]. It contains over 3600 Galactic objects classified as either confirmed ('True') PNe, Likely PNe or Possible PNe, in decreasing order of confidence. There are also about 5000 objects in the database that were originally suggested as PNe but were rejected and re-classified as a variety of different types of objects.

A notable aspect of PNe is their axi-symmetric structure. There is a wide variety of structures, seen well especially in high-resolution observations (e.g., from the Hubble Space Telescope), but they tend to fall into a few distinct groupings, namely Round, Elliptical and Bipolar morphologies. These morphologies are thought to have their origins in the envelope ejection by the originating star, where especially a binary companion may contribute to the deviations from sphericity [3]. PNe morphology has been studied since the 19th century [4], and it has grown in importance with the advances in sensitivity and resolution arising from new detector technologies and observation techniques.

The morphological classification assigned to a PN can be affected by the quality of the image. The outer regions are often faint and require high dynamic range. Many PNe that had been earlier classified as Elliptical or Round are now seen as Bipolar [5]. However, for many PNe, only images from wide-field or all-sky surveys are available, and these have limited resolution and sensitivity. For PNe close to the plane of the Galaxy, the confusion by many field stars seen near to or superposed on the nebula can also complicate the analysis of the PN image. The morphological classifications are still studied and improved upon.

In this paper, we investigate the efficacy of Deep Learning (DL) for deciding whether an object is a PN and for determining its morphological classification. We make use of the PNe images available in the HASH DB and in the Panoramic Survey Telescope and Rapid Response System Data Release 2 (Pan-STARRS) [6,7]. The main objective is to leverage knowledge from pre-trained DL models and use these to automatically distinguish True from Rejected PNe and to obtain their morphology classification. We compare various current DL models and assess their success in identifying the True PNe and determining their corresponding morphology [5]. It is a challenging problem when using typical images rather than the highest quality available for only a subset of PNe. Several related works to classify PNe have been proposed using different methods. Faundez-Abans et al. [8] performed a cluster analysis on the PN chemical composition and then trained an Artificial Neural Network using the classified chemical composition to recognize and assign the PNe into its respective type. Recently, Akras et al. [9] used a Machine Learning (ML) technique alongside the infrared photometric data to distinguish compact PNe from their mimics. Deep learning has been used for galaxy morphology classification [10], mostly utilizing the Galaxy Zoo dataset [11]. Galaxy morphologies are easier to determine, and the objects are little affected by foreground stars, as depicted in Figure 1. In contrast, PNe are more complex and are often located in dense star fields. This makes PNe a good testing case for determining the accuracy and limitations of the technique. The results can be generalized to other datasets, such as the deep-field images where the most distant galaxies also present extended objects in highly confused fields [12].

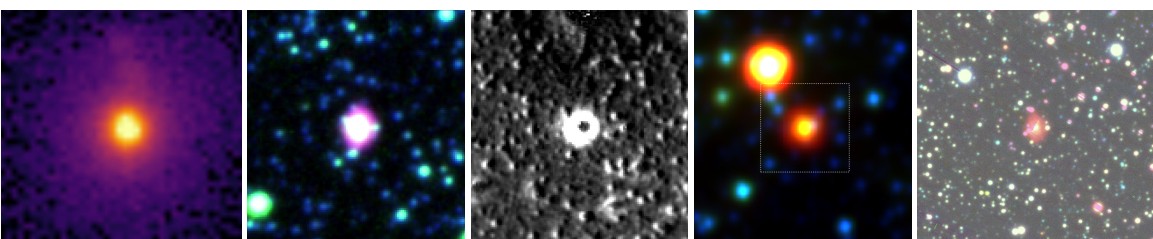

**Figure 1.** Examples images of Elliptical objects. From left to right: Elliptical galaxy from the Galaxy Zoo dataset [11]; Elliptical PNe in Optical images, H$\alpha$ "Quotient" images and infrared ("WISE432") images; and high-resolution Optical Pan-STARRS images.

Deep Learning is the emerging subdomain of ML in Artificial Intelligence (AI). The algorithm consist of deep Artificial Neural Network (ANN) layers that mimic the information-processing mechanism of the human brain. It is the state-of-the-art in computer vision and an effective image classification [13] approach as DL is capable of processing a large amount of input images without having to perform pre-processing (feature extraction, mining or engineering), and it has the capability to learn to solve complex problems without human intervention. Inspired by the nature of how humans learn by transferring and leveraging on previously obtained knowledge, we exploit the transfer learning approach. The advantages of using transfer learning is its ability to achieve faster learning processes while requiring less training time and data. The formal definition of transfer learning for this work is defined as [14]:

Given a source domain $D_s$ and its learning task $T_s$, a target domain $D_t$ and its learning task $T_t$, transfer aims to help improve the learning of the target predictive function learning $f_T(p)$ in $D_t$ from $D_s$ and $T_s$, where $D_s \neq D_t$ or $T_s \neq T_t$. $D_s$ consist the source domain data; $D_s = (x_s, y_s), ..., (x_{s_i}, y_{s_i})$, where $x_{s_i}$ is the image data instance and $y_{s_i}$ is its corresponding class label. Likewise, the target domain data $D_t = (x_t, y_t), ..., (x_{t_i}, y_{t_i})$, where $x_{t_i}$ is the input image data instance and $y_{t_i}$ is its corresponding output class label. Most often $0 < x_{t_i} \leq x_{s_i}$.

In this work, the term Deep Transfer Learning (DTL) is used to refer to the application of the transfer learning approach during the training of the DL algorithms for PNe classifications, as shown in Figure 2. The overall framework for this work is depicted in Figure 3. We first create a dataset from HASH DB and Pan-STARRS, and then select a suitable modern DL algorithm architecture to perform the transfer learning and save the best model built during training. The model is then used to classify the test images. Finally, we analyze and evaluate the results. Details of the framework components are elaborated further in the following section.

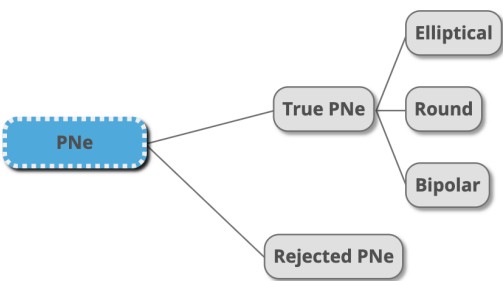

**Figure 2.** PNe classification as used in this work: True PNe versus Rejected and the three allowed morphologies of the nebulae.

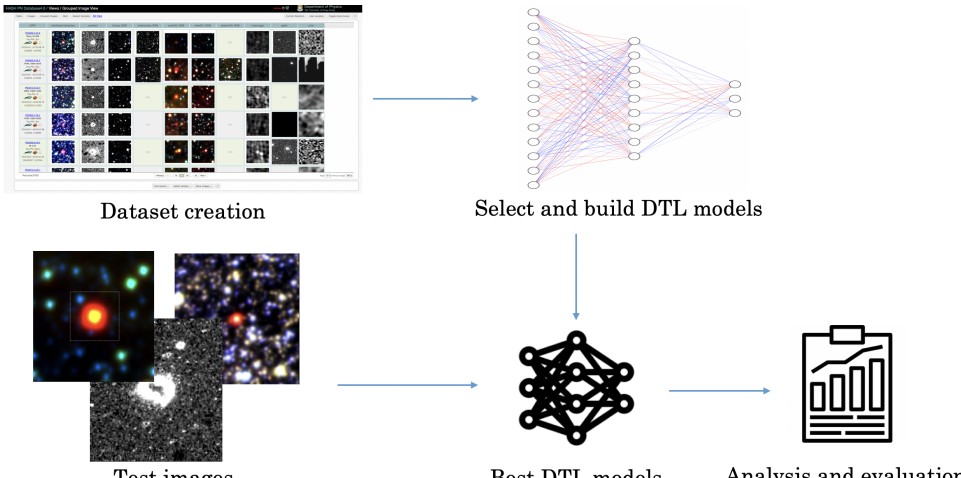

**Figure 3.** The framework for deep transfer learning for True PNe, Rejected and morphological classifications. The images shown are from the HASH DB.

## 2. Materials and Methods

### 2.1. Dataset Creation and Pre-Processing: HASH DB

To obtain the images of PNe, we used two databases, HASH DB [2] and the recent Pan-STARRS [6]. HASH DB contains a wide range of images taken with different instruments and telescopes. We selected images from wide-area surveys, mostly taken from the IPHAS/VPHAS CCD survey of the Galactic plane and the SHS/SSS photographic-plate survey at optical wavelengths, and the Wide-field Infrared Survey Explorer (WISE) all-sky survey at infrared wavelengths.

The Optical images detect the emission from the nebular gas, whilst the infrared wavelengths detect emission from small solid particles (dust) in the nebulae. Traditionally, PNe have been discovered by a combination of Optical images showing their extended morphological nature and spectroscopy detecting the bright emission lines of the nebulae (normally dominated by Hα emission near 656.3 nm). The wide-area surveys provide uniform data quality and properties for the PNe. The uniformity is a significant advantage to the DTL.

The Optical images used here are taken in several filters. The SSS and SHS surveys are photographic SuperCOSMOS Sky Surveys. SSS describes a three-band survey with broad filters (B, R and I: Hambly et al. [15]), which in HASH DB are combined into a three-color image. SHS provides images in a narrow Hα filter and a broader short-red filter [16]. HASH DB uses these to obtain a quotient image by dividing the Hα image by the continuum image. This brings out the PN while minimizing the field stars which are bright in the continuum. INT Photometric Hα Survey of the Northern Galactic Plane (IPHAS) and VST/OmegaCAM Photometric Hα Survey (VPHAS) are CCD Hα surveys of, respectively, the northern and southern halves of the Galaxy [17]. HASH DB uses the three filters employed by these surveys (r, i and Hα) to make three-color images and uses the Hα and r filters for a quotient image. For the WISE data [18], we used the '432' HASH DB image created by combining filters at 22, 11 and 4.6 μm. The IPHAS and VPHAS cover the areas within 5 degrees of the galactic plane, where most PNe are found. SHS extends to areas further from the plane. SSS is all-sky. The three-band images are hereafter called 'Optical'. Where both IPHAS/VPHAS and SSS are available, we used the former as they have better spatial resolution and dynamic range of the images.

For this research, we selected image resources that are available for the large majority of PNe. An alternative approach would have been to select images from targeted observations which are optimized for PNe. This includes observations taken with the Hubble Space Telescope. This would have given much better quality images for the DL attempted here, but with less scope for applications

as such data is typically only available for already well-studied objects. We concentrate on general surveys to test whether these methods can be used to classify less well-studied astronomical objects.

Images from the HASH DB are retrieved as PNG images which include the RA (J2000) vs. Dec (J2000) coordinate axes and labels. We automatically cropped the images to remove the white regions where the axes are located. No further image manipulation was performed, and the input images of the PNe are generally visually similar to the ones in Figure 1.

We divided the total number of images into Training (80%), Validation (10%) and Test (10%) sets. The images for the Training and Validation sets were randomly selected from all images, whereas the images for the Test set were based on selecting PNe and using all images associated with that PN. Because the Test set covers a minority of the objects, randomly selecting images for it will lead to most PNe in the test sample being represented by a single image resource only. This would not allow us to test the use of various combination of different image resources. All of these sets do not contain the same PN and images.

The Training set is used to built the DTL models. The Validation set is set aside to provide an unbiased evaluation of a model built using the Training set and to fine tune the model parameters. The Test set is used to provide an unbiased evaluation of the best model that was built on the Training set. In this work, we use the DL algorithms without tuning any parameters. Therefore, the result discussions focus on the outcome from Training and Test set, and the Validation set is only used as an intermediate check. It is worth pointing out that information about the position of the PNe in the Galaxy (which determines the density of the confusing stars in the field) and the distances of the PNe were not used in the DL training and classification. The PNe were considered as a uniform set.

*2.2. Dataset Creation and Pre-Processing: Pan-STARRS*

The HASH DB contains pre-processed images of PNe, designed to act as visual cues for human researchers. These images come from several different programs with different characteristics and include some comparatively low-resolution photographic datasets. Ideally, a comprehensive dataset should provide uniform resolution; cover a large portion of the sky; be sufficiently deep that faint, extended emission from the outer regions of nebulae is recovered; and contain a sufficiently wide set of color information that an emission spectrum can be distinguished from blackbody emission in the color data. These criteria are currently best met in the Pan-STARRS survey, which contains five-color (*grizy*-band) images of roughly three-quarters of the sky at arcsecond resolution, where the *r*-band includes the H$\alpha$ emission line. Its main draw-back is that it lacks a narrow-band H$\alpha$ filter which would have increased sensitivity to PNe. Pan-STARRS images are currently not included in the HASH DB. We added these data separately to our image resources.

To obtain a uniform dataset of PNe, we use the Pan-STARRS image cutout API[1] to extract $600 \times 600$ pixel FITS images in each filter, centred on the co-ordinates of the object as listed in HASH. Of the 3617 objects in the HASH DB, 2356 have a complete set of *grizy* observations.

To produce color images from these, each FITS image was clipped to remove the brightest 2.5% of pixels (set as white) and combined so that the blue, green and red channels ($B, G, R$) of the final image were represented by

$$
\begin{aligned}
B &= g + r/2, \\
G &= r/2 + i + z/2, \\
R &= z/2 + y.
\end{aligned}
\tag{1}
$$

---

[1]   https://ps1images.stsci.edu/ps1image.html.

Each of these three channels were then normalized on an 8-bit scale (0–255) to produce a color image. This was cropped to 512 × 512 pixels, and then scaled to the stated input size for the relevant DL algorithm. These are hereafter referred to as the 'plain' set of images.

Many known or suspected PNe are located in the Galactic bulge and Galactic plane, where stellar densities are high. Frequently, they are among the fainter objects in the surrounding projected field and are often lost in the glare of many brighter stars. To try to circumvent this, two further sets of images were produced: one where an effort was made to remove foreground and background stars from the images (referred to as 'No-star' images) and one where a mask was generated to remove emission from any sources other than the PN (hereafter 'Mask'). The additional processing steps to create these alternate images were performed on the original FITS images before clipping.

Best-practice for removal of stars from images generally involves calculating and removing a point-spread function (PSF) for each star (e.g., [19]). However, characterizing an accurate PSF for each observation and in each filter and ensuring its creation and subtraction while accounting for non-linearity, saturation and background correction are too complex an endeavor to be attempted here. Instead, we take the approach of treating stellar PSFs as bad data and masking them from the image, either using a median filter for the 'No-Star' images or an image mask.

Stars were identified for removal by searching for local maxima in the images, within a certain neighborhood radius. Data were then median-filtered on the same radius, and this median-filtered image was subtracted, leaving a high-pass-filtered image showing small-scale structure. If the brightness of a star in this high-pass image exceeded a threshold, it was flagged for removal. Stars within $\sqrt{2}$ times the neighborhood radius of the PN centre were ignored, in order to avoid masking the central star of the nebula.

Many stars lie within the nebula emission itself. Thus, it is important to mask out no more of the image than necessary. For the 'No-Star' images, annuli were drawn around the star in the original image at one-pixel intervals, and the median value in each annulus was calculated. The reduction in median flux between one annulus and the next was calculated, and annuli were flagged for replacement if that reduction exceeded a tolerance (out to a certain maximum radius).

The replacement value used was the median of the next annulus from the star (i.e., the median flux of the first annulus not to show a substantial reduction in median flux with radius, $r$, denoted $M$ in the following). However, this had a tendency to produce faint, round 'ghost' stars in the images (Figure 4). Consequently, a hardness parameter ($h$) was introduced, which allowed for a weighted mean of this median and the original data ($D$) to generate the replacement dataset ($D'$), namely

$$D' = fD + (1 - f)M, \tag{2}$$

where

$$f = \left(\frac{r}{R}\right)^h \tag{3}$$

and $R$ is the maximum allowed radius for removal. This both avoids hard edges to the removed regions and allows $R$ to be expanded to larger radii without greatly affecting the nebula.

This procedure was repeated four times to remove progressively fainter stars, using different parameters in each iteration. Through trial and error, we determined an appropriate set of parameters for the Pan-STARRS dataset: a neighborhood size of 15, 13, 11 and 9 pixels; $R = 25, 20, 15$ and 10 pixels; tolerances of factors of 1.01, 1.02, 1.03 and 1.04; thresholds of 0.3%, 0.7%, 1.0% and 2.5% of the image's brightest pixels; and $h = 7, 5, 3$ and 1, for the four iterations, respectively. One pixel in Pan-STARRS correspond to 0.26 arcsec.

Visual inspection of the images with stars removed in this manner showed that it was effective in ensuring the fainter, diffuse emission from the nebula was given more prominence in the images. However, the removal of stars was still imperfect, and a large number of objects classified as True PNe remained as a relatively faint, unresolved source in the centre of the images.

An attempt was then made to generate a mask around the emission from the central source. This began by filtering the star-subtracted data with a 21-pixel-radius median filter. The median flux of this filtered image was subtracted to leave an image showing only large-scale structure and with an overall median flux of zero. We ordered the pixels in this image by flux and calculated the 2.5th percentile flux as a benchmark flux. Working on the assumption that the majority of the image remains Gaussian-distributed noise, the negative of this benchmark should approximate the $2\sigma$ upper bound (97.5th percentile) of noise in the data, and any greater flux should represent emission from the PN (or surrounding stars). A mask was generated such that any areas of emission that were contiguous with that of the central star remained in the image, and the remainder of the image was set to black. The mask was not applied to each filter, but to the overall image, with areas passed by the mask if at least three of the five bands showed emission. In practice, this masking process was not very effective for many PNe. As shown in Figure 4, it proved difficult to identify an appropriate cut-off percentile that satisfied both the need to remove overlapping PSFs of unrelated stars, and the need to retain faint emission from the edges of the PNe.

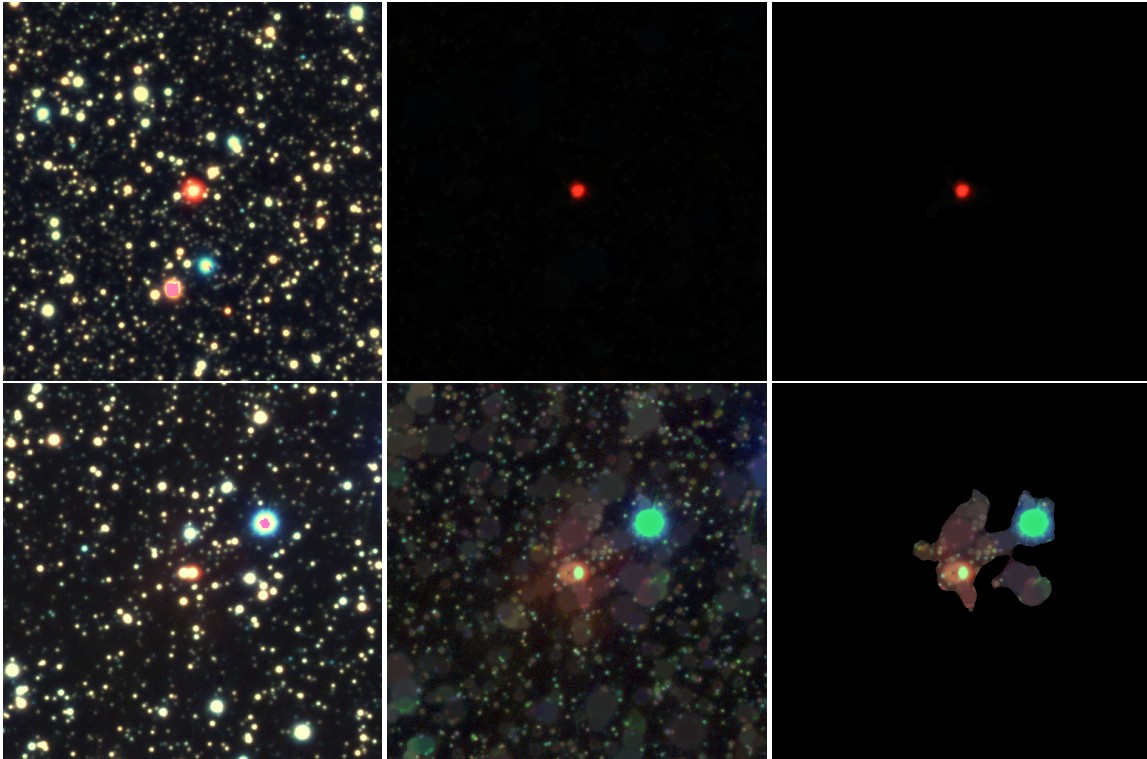

**Figure 4.** Examples of PNe from the Pan-STARRS survey (**left**), showing (**top**) successful and (**bottom**) unsuccessful algorithmic removal (**middle**) and masking (**right**) of contaminating stars. The bottom example shows the difficulties in isolating the faint nebular emission (the diffuse red glow in the bottom-centre panel) from the dense field of background stars.

To reduce the confusion generated by foreground and background stars, as well as by the large number of PNe that remained as point sources in the images, a subset of objects were selected from the HASH DB that are at least $2''$ in diameter along their major axis and lie at least $2°$ from the Galactic plane.

2.2.1. Sample Selection for True PNe and Rejected Classification

We queried the HASH DB using the 'select sample' option provided by HASH DB in the combined search user interface. We submitted the query listed in Table A1 for retrieving the True PNe, the Rejected PNe and other objects. The True PNe have been confirmed as PN while the latter are considered not to

be PN. The HASH DB contains separate lists of objects suspected to be PNe but not yet confirmed as such: these are listed as Likely PNe or Possible PNe, depending on the degree of confidence. 'Likely' PNe have a higher degree of confidence and a PN is the most likely classification, but lack confirmation from spectra or images; 'Possible' PNe have inconclusive spectra and images and PN classification is one of several possibilities [2]. We also retrieved these.

To select the Rejected PNe and other objects, we searched for the following object types in the HASH DB: AGB star, AGB star candidate, artifact, Be star, cataclysmic variable star, circumstellar matter, cluster of stars, cometary globule, emission-line star, emission object, galaxy, Herbig–Haro Object, H II region, interesting object, ionized ISM, object of unknown nature, objects to check, PAGB/Pre-PN (post-AGB stars), possible Be star, possible emission-line star, possible galaxy, possible Herbig–Haro Object, possible pre-PN, possible transient event, RCrB/eHe/LTP (post-PNe objects), reflection nebula, RV Tau, star, supernova remnant, supernova-remnant candidate, symbiotic star, symbiotic star candidate, test object, transient event, transition object, white dwarf/hot sub-dwarf, young stellar object and young stellar object candidate.

During the period of our data collection (April 2020), the total number of True PNe returned by HASH DB was 2450. The distribution details of the True PNe according to their image resources are listed in Table 1. Based on the total number images for each type of image resources, we decided to use 2100 images from the HASH DB as samples for True PNe and Rejected classes. This is because of two factors: first, the number of Quotient images available; and, second, that our approach is to perform DTL on a balanced dataset. All of the Possible and Likely PNe images were used to predict whether they fall into True PNe or Rejected class. For this test, we only use the Plain images from the Pan-STARRS survey: the total number of Pan-STARRS images of True PNe is 1508 and that of Rejected class is 1768, and we used 1500 images from each class for the DL algorithms. The details of the distribution for the True PNe and Rejected class from HASH DB and Pan-STARRS used for the DL algorithms are in Table 2.

**Table 1.** Distribution of True PNe, Rejected PNe/Other Objects, Possible PNe and Likely PNe alongside their respective image resources from the HASH DB.

| Class | Total # PNe | Total # Images | Optical | Quotient | WISE432 | Pan-STARRS |
|---|---|---|---|---|---|---|
| True PNe | 2450 | 17,612 | 2443 | 2101 | 2441 | 1508 |
| Rejected PNe/Other Objects | 2741 | 18,507 | 2696 | 2159 | 2694 | 1768 |
| Possible PNe | 368 | 2630 | 367 | 330 | 368 | 216 |
| Likely PNe | 313 | 2287 | 311 | 282 | 312 | 242 |
| Grand Total | 5872 | 41,036 | 5817 | 4872 | 5815 | 3734 |

**Table 2.** Dataset distribution for True and Rejected PNe from the HASH DB and Pan-STARRS.

| Dataset | Percentage HASH DB/Pan-STARSS | HASH DB Number of Images | Pan-STARRS Number of Images |
|---|---|---|---|
| Training | 80%/77% | 1680 | 1200 |
| Validation | 10%/10% | 210 | 150 |
| Test | 10%/13% | 210 | 210 |
| Total number of images for each image resource | | 2100 | 1560 |
| Total number of images for each PNe class | | 6300 | 1560 |
| Total number of images used for True and Rejected PNe Classification | | 12,600 | 3120 |

### 2.2.2. Sample Selection for PNe Morphological Classification

The morphological classification of the PNe in HASH DB is based on Corradi and Schwarz [20,21]. The retrieved images returned by the True PNe query (in Section 2.2.1) were downloaded and consolidated as a collection of PNe based on their morphologies and type of image resources. Distribution details of the True PNe morphologies and image resources from HASH DB and

Pan-STARRS are in Table 3. The number of images for each different type of Pan-STARRS image resources (Plain, Quotient, No-star and Mask images) are the same as the number of PNe.

Since our approach is to create a DL model out of a balanced image distribution, we selected the three most frequent morphologies (Bipolar, Elliptical/Oval and Round), which have a reasonable number of examples to learn from. The Asymmetric and Irregular classes have too few objects. The Quasi-stellar class refers to unresolved PNe for which no morphological information is available. In total, 280 images from each type of HASH DB PNe image resources were randomly selected as examples for the three morphologies (hence the total of 840 images). Choosing 280 images allows us to later build a model that comprises of True PNe, Likely PNe and Possible PNe. As for images from Pan-STARRS, we used 160 images for each type of morphology, set by the limit of the samples for Bipolar images. Details of the distribution are tabulated in Table 4.

**Table 3.** Distribution of Morphology and the image resources from HASH DB.

| Morphology | Total Number of PNe | Total Number of Images | Optical | Quotient | WISE432 | Pan-STARRS |
|---|---|---|---|---|---|---|
| Asymmetric | 9 | 69 | 9 | 8 | 9 | N/A |
| Bipolar | 543 | 3857 | 542 | 464 | 540 | 161 |
| Elliptical/oval | 1017 | 9764 | 1010 | 861 | 1012 | 390 |
| Irregular | 18 | 135 | 18 | 15 | 18 | N/A |
| Quasi-Stellar | 374 | 2829 | 370 | 350 | 372 | N/A |
| Round | 489 | 3408 | 489 | 397 | 487 | 200 |
| Grand Total | 2450 | 20,062 | 2438 | 2095 | 2438 | 751 |

**Table 4.** Dataset distribution for each type of morphology from HASH DB and Pan-STARRS.

| Dataset | Percentage | HASH DB Number of Images | Pan-STARRS Number of Images |
|---|---|---|---|
| Training | 80% | 224 | 128 |
| Validation | 10% | 28 | 16 |
| Test | 10% | 28 | 16 |
| Total number of images for each morphology | | 280 | 160 |
| Total number of images for each image resource | | 840 | 640 |
| Total number of images used for PNe Morphology Classification | | 2520 | 1920 |

### 2.3. Deep Transfer Learning Algorithm Selection

Instead of initiating a new DL process to learn the PNe classification and morphological structures from scratch, we applied transfer learning from existing popular DL algorithms. These algorithms were trained to learn a large-scale image-classification task according to the visual categories in the ImageNet dataset, which contains 14 million images corresponding to 22 thousand visual categories [22]. For an initial study, we selected eight DL algorithms for which the classification effectiveness was validated using ImageNet by Keras [23]. We found that the three selected DL algorithms in Table 5 were the most effective in classifying PNe. AlexNet [24], VGG-16 [25], VGG-19 [25], ResNet50 [26] and NASNetMobile [27] were also tested but were found to be less effective and were dropped from further consideration. The algorithms and the effectiveness from [23] are listed in Table 5.

Many efforts have been made to improve the original architectural design of Convolutional networks (ConvNets) to achieve better accuracy. One of the improvements dealt with the depth of the ConvNets where other parameters of the architecture are fixed while the depth of the network is steadily increased by adding more convolutional layers. An example is the Inception architecture, which achieved very good performance at a relatively low computational time. Residual networks (ResNets) [26] have been introduced to address the limitation of computational time required to train very deep ConvNets. Recently, the introduction of residual connections along with traditional architecture has yielded state-of-the-art performance. Inception-ResNet-v2 is the combination of the Inception architecture and residual connections. This idea was studied by Szegedy et al. [28] and

the experimental results clearly show that the speed of training the inception networks with residual connections (Inception-ResNet-v2) has been significantly improved.

**Table 5.** List of DL algorithms used in this work for PNe True versus Rejected and for the morphological classification.

| Selected DL Algorithms | Top-1 Accuracy [23] |
| --- | --- |
| InceptionResNetV2 (2016) [28] | 0.803 |
| DenseNet201 (2017) [29] | 0.773 |
| ResNet50 (2015) [26] | 0.749 |
| NASNetMobile (2017) [27] | 0.744 |
| VGG-16 (2105) [25] | 0.713 |
| VGG-19 (2105) [25] | 0.713 |
| MobileNetV2 (2018) [30] | 0.713 |
| AlexNet (2012) [24] | 0.633 |

Another DL architecture inspired by ResNets is DenseNet [29]. DenseNet has proven to utilize significantly fewer parameters and less computation by introducing a computational approach that leads to shorter connections in between the early layers and later layers. This approach had resulted in input feature-maps that are reusable, accessible to all layers in the network and closer to the output layer. DenseNet comes in various network architectures; DenseNet with 201 layers (known as DenseNet-201) has been shown to be the most effective among the variations.

MobileNetV2 was introduced by Google [30]. In its original version—MobileNetV1, a *Depthwise Separable Convolution* was introduced which dramatically reduced the complexity cost and model size of the network. As the name implies, MobileNet is suitable for applications using mobile devices, or any devices with low computational power. The DL network architecture consist of two main layers. The first layer is known as depthwise convolution; it performs lightweight filtering by applying a single convolutional filter per input channel. The second layer is a $1 \times 1$ convolution, known as pointwise convolution, which is responsible for building new features through computing linear combinations of the input channels. ReLU6 is used due to its robustness when used with low-precision computation. In the second version of MobileNet (MobileNetV2), a better module was introduced with an inverted residual structure and the non-linearities in narrow layers were removed. MobileNetV2 is one of the best DL algorithm for feature extraction; it has achieved state-of-the-art performance using ImageNet. It is also widely used for object detection and semantic segmentation.

There are two popular DTL strategies. The first is using a pre-trained model as feature extractor and leverage on the pre-trained model's weighted layers to extract features but not to update the weights of the model's layers during training with new data. The second strategy is to perform fine tuning using the pre-trained model by updating the hyper-parameters and model-parameters of selected layers in the network. A DL training process involves tuning hyper-parameters and model-parameters. Hyper-parameters are the DL architecture properties that govern the entire training process, which are set before the training process starts such as the number of epochs, learning rate, hidden layers, hidden units and activation functions. The model parameters are values estimated based on the training data, which are the weights and bias in the DL architecture. The process of finding the best hyper-parameters and model parameters requires expertise and extensive trial and error. Frozen parameters are parameter values that are not changed or updated. As this is an initial study on how DL can be used to classify PNe, we focus the evaluation on the effectiveness of different DL models. Nevertheless, we executed several preliminary experiments that automatically update the model weights based on the training data, and the results are better than the model with frozen parameters.

By taking advantage of the learned feature maps from the selected pre-trained DL algorithms, we adopted the first DTL strategy to extract meaningful features from the PNe images. Transfer learning was done for all the selected pre-trained algorithms using the same approach: A new DTL model was composed by loading the selected pre-trained DL algorithms (as the convolutional base)

with ImageNet weights as the initial starting weight and stacking a classification layer on top to represent the PNe class and morphologies as output (depicted in Figure 5). All of the selected pre-trained DL algorithms were used without the original classification layers and the convolutional parameters were frozen and used as the feature extractor. As the PNe class and morphology classifier, we used a global average pooling layer and fed its output directly into the softmax activated layer. For algorithms where the output is a raw prediction value (logit), we omitted the softmax activation layer and replaced it with a Dense layer.

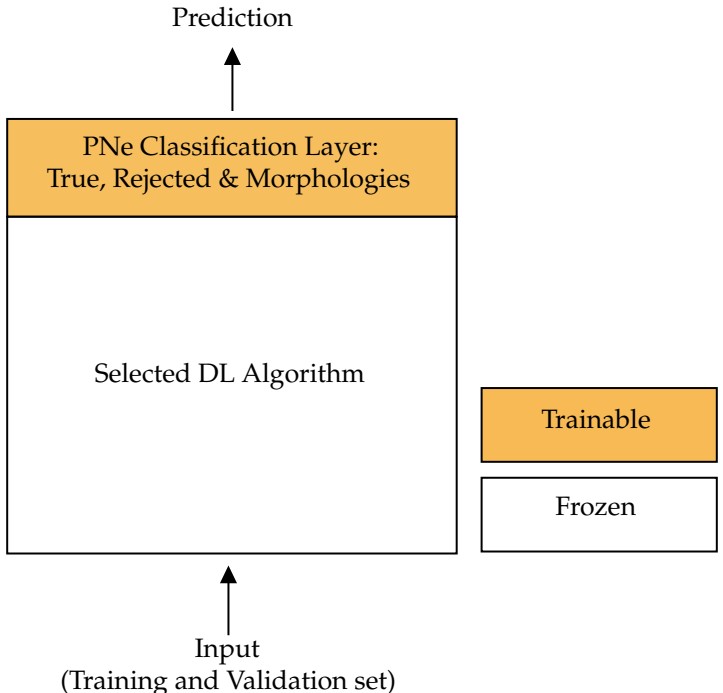

**Figure 5.** Conceptual view of the Deep Transfer Learning (DTL) Architecture used in this work.

In this work, our initial experimental strategy was executed in Python 3.7 using Google Colab's GPU [31]. The DTL model was implemented using TensorFlow version 2.0 [32] and Keras applications modules [23]. Keras with TensorFlow backend `evaluate()` function was used to evaluate the fit performance of the built model and the `predict()` function was used to predict the images in the Test set. However, the `evaluate()` and `predict()` functions produced inconsistent results. The output of the `predict()` function could not be reconciled with the success rate returned by the `evaluate()` function and was considered suspect. To address the issue, we used our own local GPU server (Tesla V100, 16 GB, 5120 CUDA Cores and 640 Tensor Cores used for DL computations) and MATLAB to execute the same DTL models and evaluation. This produced internally consistent results. All of the DTL models are trained and build using the same hyper-parameters of 64 batches; Root Mean Square Propagation (RMSprop) with the learning rate of 0.0001 as the loss optimizer; and 100 epochs. The model-parameters remained unchanged except for the last layer that was used to classify the PNe class. The model-parameter details for the DTL are as in Table 6. STEM is the number of parameters when a model is trained without the original classification layers.

**Table 6.** Model and parameter details.

| Model | Image Size | STEM |
|---|---|---|
| InceptionResNetV2 | 299 | Total parameters: 66,920,163<br>Trainable parameters: 66,859,619<br>Non-trainable parameters: 60,544 |
| DenseNet201 | 224 | Total parameters: 30,364,739<br>Trainable parameters: 30,135,683<br>Non-trainable parameters: 229,056 |
| MobileNetV2 | 224 | Total parameters: 2,260,546<br>Trainable parameters: 2562<br>Non-trainable parameters: 2,257,984 |

*2.4. Evaluation Metrics*

A binary classification was used for True versus Rejected PNe. In contrast, multi-class classification was used to classify the PNe into their respective morphology, where an object is assigned to only one class out of $n$ distinct classes [33]—in this case, Bipolar, Elliptical or Round. We evaluated the effectiveness of using the DTL models using accuracy, F1 score and two other of the most commonly used evaluation metrics based on relevance judgement, namely precision and recall [34]. Accuracy is how often the DTL model correctly classifies the PNe into its correct class. F1 score is the harmonic mean of the precision and recall. Based on the confusion matrix in Figure 6, the formal definition for all the evaluation metrics are:

$$\text{Accuracy} = \frac{(T_p + T_n)}{(T_p + F_p + F_n + T_n)} \tag{4}$$

$$\text{Precision} = \frac{T_p}{(T_p + F_p)} \tag{5}$$

$$\text{Recall} = \frac{T_p}{T_p + F_n} \tag{6}$$

$$\text{F1 Score} = \frac{2 \times (Recall \times Precision)}{(Recall + Precision)} \tag{7}$$

| | Predicted | |
|---|---|---|
| | Positive | Negative |
| **Actual** Positive | True Positive ($T_p$) | False Negative ($F_n$) |
| **Actual** Negative | False Positive ($F_p$) | True Negative ($T_n$) |

**Figure 6.** Confusion matrix for the evaluation measures.

**3. Results**

In this section, we provide the experimental findings of the DL algorithms in classifying PNe as True versus Rejected and for PNe morphologies. The highlight of these results are the effectiveness

of using the three DL algorithms to classify PNe in their categories and the outcome of predicting whether Possible and Likely PNe are True or Rejected.

We describe the results obtained using the Training and Test sets. As InceptionResNetV2 requires a very high computation load and is time consuming, we did not manage to execute it on our local GPU server. Hence, the results presented for this model are obtained from the initial experiments using the Google Colab's GPU. As for DenseNet201 and MobileNetV2, the results are from our local GPU server. The results are still comparable since the presented results for InceptionResNetV2 use the `evaluate()` function which in our evaluation worked correctly. All the tabulated results are interpreted in the following manner: the most effective result among all image resources and DTL models are bold and the best results among the DTL models for each type of image resource are underlined. Note that in some cases the differences are within the statistical (stochastic) noise.

### 3.1. Planetary Nebulae True vs. Rejected Classification

The evaluation outcome of the expected performance of the DTL model built during the training process is shown in Table 7. Based on the comparison of using four different image resources and the three DTL models for classifying True PNe versus Rejected, the highest accuracy was achieved by DenseNet201 with the Quotient images using the Training and Test set. DenseNet201 was also the best DTL model when using Optical (achieved highest accuracy, precision and recall) and WISE432 (achieved highest accuracy and recall) images. Using Pan-STARRS Plain images, MobileNetV2 achieved highest accuracy and precision, and both DenseNet201 and MobileNetV2 achieved the highest Recall. Averaged over all categories, DenseNet201 achieves slightly higher score (82%) than MobileNetV2 (81%), but the difference is not significant.

A further investigation was conducted to analyze the expected classification effectiveness of each of the classes using the Test set shown in Figure 7. We found that the average F1 score of the Optical images for True PNe and Rejected classification was 81%. The classification of True PNe and Rejected classes using Quotient images was consistently high across all DTL models. For WISE432 and Pan-STARRS, InceptionResNetV2 returned F1 scores that were higher for the Rejected class than for True PNe. For Optical and Quotient images, the F1 scores are similar for the two classes across all DTL models. The DenseNet201 model yielded the most effective algorithm with the average F1 score of 82% for both classes.

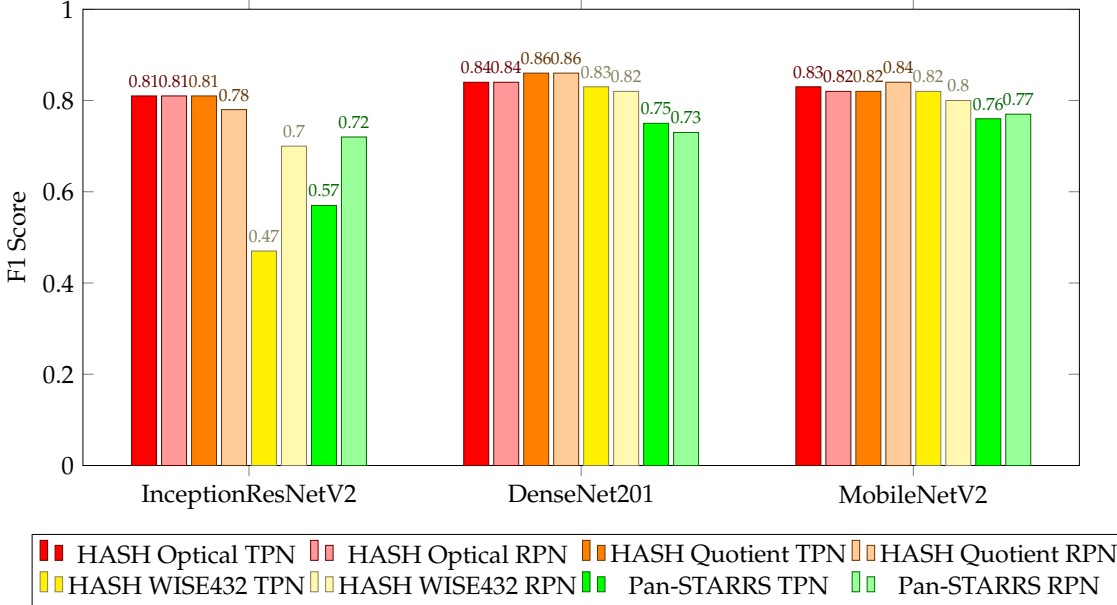

**Figure 7.** The trained model evaluation F1 Score for PNe True and Rejected classification using images from HASH DB and Pan-STARRS Test set.

**Table 7.** The trained model evaluation results for True and Rejected PNe Classification from HASH DB and Pan-STARRS. The values are the average accuracy, precision and recall for both True PNe and Rejected classes.

| | DTL Models | Training Set Accuracy | Accuracy | Test Set Precision | Recall |
|---|---|---|---|---|---|
| **HASH Optical** | InceptionResNetV2 | 0.80 | 0.81 | 0.78 | 0.78 |
| | DenseNet201 | <u>0.86</u> | <u>0.84</u> | <u>0.85</u> | <u>0.82</u> |
| | MobileNetV2 | 0.83 | 0.83 | 0.83 | <u>0.82</u> |
| **HASH Quotient** | InceptionResNetV2 | 0.77 | 0.80 | 0.81 | 0.75 |
| | DenseNet201 | **0.88** | **0.86** | **0.88** | 0.84 |
| | MobileNetV2 | 0.84 | 0.83 | 0.79 | **0.86** |
| **HASH WISE432** | InceptionResNetV2 | 0.62 | 0.61 | 0.61 | 0.62 |
| | DenseNet201 | 0.81 | <u>0.82</u> | 0.81 | <u>0.85</u> |
| | MobileNetV2 | <u>0.84</u> | 0.81 | <u>0.86</u> | 0.78 |
| **Pan-STARRS Plain** | InceptionResNetV2 | 0.62 | 0.66 | 0.66 | 0.63 |
| | DenseNet201 | <u>0.81</u> | 0.74 | 0.72 | <u>0.78</u> |
| | MobileNetV2 | 0.77 | <u>0.76</u> | <u>0.74</u> | <u>0.78</u> |

The most effective result among all image resources and DTL models are bold and the best results among the DTL models for each type of image resource are underlined.

## 3.2. Prediction

As DenseNet201 was evaluated to be the top-scoring DTL model, we focused on this implementation for the next step. We predicted whether a particular object is a PN using the Test set for each of the available images (Optical, Quotient, WISE432 and Pan-STARRS Plain). The four predictions were combined with equal weights to produce a predicted final class for a particular planetary nebula. A similar method of using several diagnostics and averaging the DL outcomes was used by Zhu et al. [35].

The results are presented as a confusion matrix. When a particular planetary nebula falls into both classes with equal classification probability, we excluded that particular planetary nebula from the confusion matrix. Figure 8 shows the combined classification probability of 210 planetary nebula and 210 other objects in the Test set, which are then used to derive the confusion matrix. The total number of planetary nebula/other objects that can be confidently classified (combined classification probability $\neq$ 50%) is 347. The results show that True PNe are correctly classified in 94% of cases (precision = 0.94).

The Matthews correlation coefficient of the confusion matrix, $\phi$, provides an unbiased metric for the performance when the categories do not have equal sizes. The metric runs from $-1$ to $+1$, where 0 indicates a random result and 1 is a perfect classification. We found $\phi = 0.90$, indicating a good performance.

For further evaluation, we reduced the classification weight of Pan-STARRS (as it has the lowest accuracy). This prevents the inconclusive possibility of 50% probability, and thus allows all the PNe to be classified as either True PN or Rejected. The number of PNe correctly classified as True PNe increased from 179 to 190, leaving 20 True PNe classified as Rejected. On the other hand, the number of PNe correctly classified as Rejected increased from 168 to 195, leaving 15 Rejected PNe classified as True PNe. This reduces $\phi$ to 0.83. Down weighting Pan-STARRS for the inconclusive objects does not improve the classification confidence.

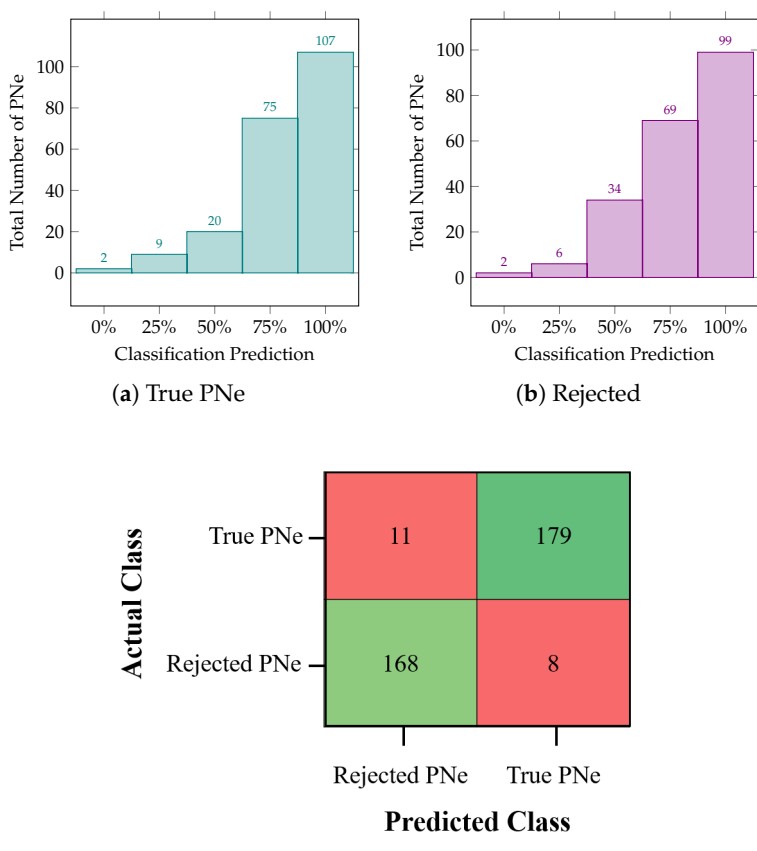

(**a**) True PNe    (**b**) Rejected

(**c**) Confusion matrix

**Figure 8.** Combined predictions for True PNe and Rejected class using the DenseNet201 DTL model: (**a**) probability distribution histogram for True PNe prediction; (**b**) probability distribution histogram for Rejected prediction; and (**c**) confusion matrix of the combined predictions derived from (**a**,**b**).

Figure 9 shows the classification probability for the resources separately. The black lines shows the correctly classified objects and the red line where the assigned classification disagrees with that in the catalog. Optical, Quotient and WISE432 all show a high degree of confidence, with probability for the correctly classified objects peaking at over 90%, for both the True and Rejected PNe. Pan-STARRS also shows a good result but the probabilities are not quite as high. This is understandable because Pan-STARRS lacks a filter dedicated to the emission lines that are characteristic for PNe.

### 3.3. Possible and Likely Planetary Nebulae Classification

The applicability of these DTL models results were then tested on the Possible and Likely PNe. The same approach was used to create the confusion matrix: each image resource was used separately to classify each PN into True PNe or Rejected classes, and subsequently all image resources for each PN were combined to arrive at a classification. From a total of 681 Possible and Likely PNe, we were able to classify 578 PNe into either True PNe or Rejected PNe. As depicted in Figure 10, the Likely PNe are classified as True in 64% of 260 classifiable cases (out of 313 in total), while for Possible PN this fraction is 41% of 318 classifiable objects (out of 368 in total).

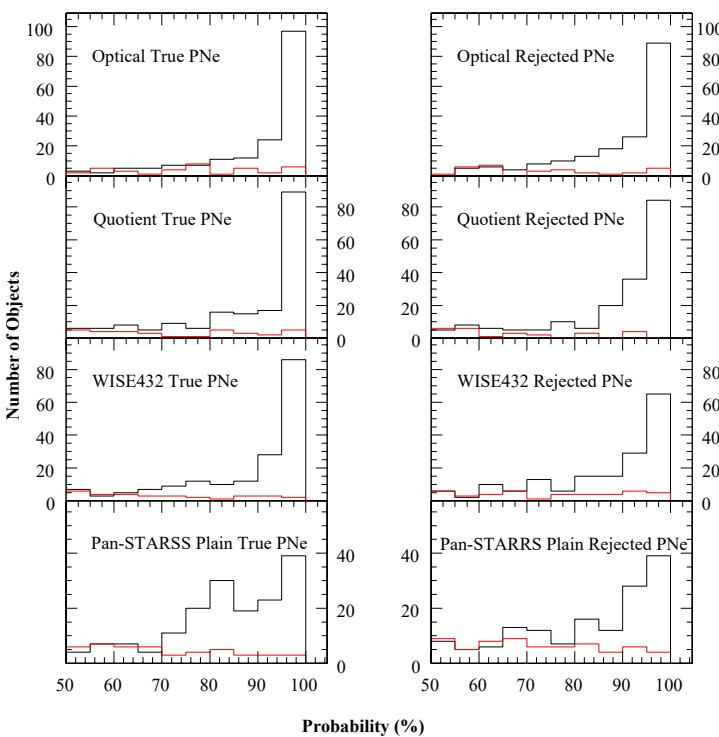

**Figure 9.** The histogram of the probabilities assigned by DenseNet201 DTL model to the PNe in the HASH Optical Test set. The *x*-axis shows the probability score assigned a PN to both classes. The *y*-axis shows number of objects per bin. The plots on the left show the True PNe and on the right the Rejected PNe. Black lines show correctly classified (true positives on the right, true negatives on the left) and red lines show the misclassified objects (false negatives on the left, false positives on the right).

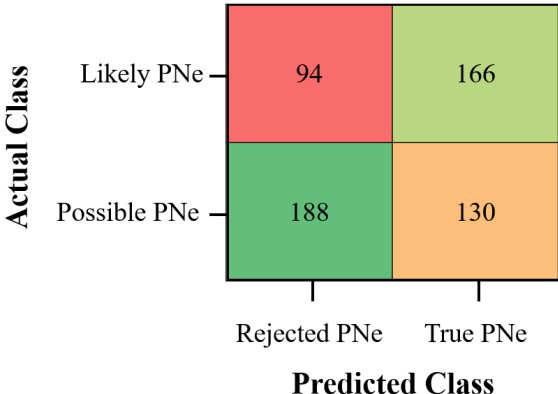

**Figure 10.** The confusion matrix of combined DenseNet201 DTL models predictions for Possible and Likely PNe.

The higher success rate for Likely PNe agrees with the original level of confidence which is higher for 'Likely PN' than for 'Possible PN'. The DTL indicates that the majority of Likely PNe are indeed PN, but that for the Possible PNe, the majority are not.

*3.4. Planetary Nebulae Morphology Classification*

In this section, we discuss the evaluation outcome of the built model for PNe morphology classification. We first start with the overall results of InceptionResNetV2, DenseNet201 and MobileNetV2 for image from HASH DB and Pan-STARRS. The results are compared between the Training and Test sets. Then, we present our findings for the classification of the Bipolar, Elliptical and Round morphologies.

The PNe morphology classification was carried out using images of the True PNe from HASH and Pan-STARRS. We experimented using seven type of image resources: Optical, Quotient and WISE432 from HASH DB and Plain, Quotient, No-star and Mask images that were derived from Pan-STARRS. Based on Table 8, the results from training DTL models using InceptionResNetV2 produced models with 100% accuracy. However, the Test set does not achieve this: when comparing the training results to those of the test results, the highest average accuracy, precision and recall in classifying the three type of PNe morphologies was 71% by using MobileNetV2 with the Pan-STARRS Plain images. We acknowledge the possibility of overfitting when 100% accuracy was obtained using InceptionResNetV2 (executed using TensorFlow and Keras), and the Test set gives a better indication of the success rate.

**Table 8.** Average accuracy, precision and recall for planetary nebulae morphology classification using image resources from HASH DB and Pan-STARRS.

| | DTL Models | Training Set Accuracy | Accuracy | Test Set Precision | Recall |
|---|---|---|---|---|---|
| **HASH Optical** | InceptionResNetV2 | **1.00** | 0.15 | 0.17 | 0.15 |
| | DenseNet201 | 0.93 | <u>0.70</u> | 0.54 | <u>0.55</u> |
| | MobileNetV2 | 0.86 | <u>0.70</u> | <u>0.56</u> | <u>0.55</u> |
| **HASH Quotient** | InceptionResNetV2 | 0.86 | 0.47 | <u>0.46</u> | 0.39 |
| | DenseNet201 | <u>0.91</u> | <u>0.63</u> | 0.45 | <u>0.44</u> |
| | MobileNetV2 | 0.86 | 0.52 | 0.30 | 0.28 |
| **HASH WISE432** | InceptionResNetV2 | 0.41 | 0.34 | 0.37 | 0.30 |
| | DenseNet201 | <u>0.95</u> | <u>0.64</u> | <u>0.47</u> | <u>0.45</u> |
| | MobileNetV2 | 0.86 | 0.59 | 0.38 | 0.38 |
| **Pan-STARRS Plain** | InceptionResNetV2 | **1.00** | 0.48 | 0.49 | 0.44 |
| | DenseNet201 | 0.97 | 0.58 | 0.37 | 0.38 |
| | MobileNetV2 | 0.98 | **0.71** | **0.59** | **0.56** |
| **Pan-STARRS Quotient** | InceptionResNetV2 | **1.00** | 0.38 | <u>0.45</u> | <u>0.39</u> |
| | DenseNet201 | 0.98 | <u>0.55</u> | 0.32 | 0.34 |
| | MobileNetV2 | 0.90 | 0.54 | 0.30 | 0.31 |
| **Pan-STARRS No-star** | InceptionResNetV2 | 0.97 | 0.38 | 0.40 | 0.39 |
| | DenseNet201 | <u>0.98</u> | <u>0.63</u> | <u>0.44</u> | <u>0.44</u> |
| | MobileNetV2 | 0.96 | 0.61 | 0.42 | 0.42 |
| **Pan-STARRS Mask** | InceptionResNetV2 | 0.84 | 0.38 | 0.39 | 0.31 |
| | DenseNet201 | <u>0.98</u> | <u>0.65</u> | <u>0.47</u> | <u>0.47</u> |
| | MobileNetV2 | 0.82 | 0.59 | 0.38 | 0.39 |

The most effective result among all image resources and DTL models are bold and the best results among the DTL models for each type of image resource are underlined.

Comparing the model evaluation outcome obtained using Pan-STARRS images, we found that MobileNetV2 was the best overall performing DTL model. As Pan-STARRS Plain images can be considered the same type of image resource as HASH Optical, the results confirm that images from Pan-STARRS can be a good alternative. However, HASH Quotient images performed better than Pan-STARRS Quotient images, possibly because the Pan-STARRS filters are not optimized for the PN emission lines.

We used four different Pan-STARRS resources. The three additional resources experimented with different ways to address the star-nebula confusion. However, this did not result in a notable improvement.

Classification Accuracy of Bipolar, Round and Elliptical Planetary Nebulae

From this section onward, we focus the discussion on the classification of PNe morphologies based on the F1 scores for the images from the HASH DB and Pan-STARRS Test set.

The results for Bipolar PNe classification depicted in Figure 11 demonstrate that InceptionResNetV2 was the most effective DTL model for classifying the Bipolar PNe, with the average F1 score of 56%, followed by MobileNetV2 with the average F1 score of 49% and DenseNet201 with the average F1 score of 47%. This hides large variations. All three models did reasonably well on HASH Optical images. DenseNet201 was best for the HASH Optical but worse for Pan-STARRS Plain. MobileNetV2 was more consistent but failed at the Pan-STARRS resources except for Pan-STARRS Plain and Mask. InceptionResNetV2 was more consistent, and it was notably the only routine able to handle the Pan-STARRS No-star images. DenseNet201 was not able to classify the Pan-STARRS Plain Bipolar PNe images: out of the 16 test images, only three were correctly classified, and the majority of the remainder were classified as Round PNe.

Figure 12 shows the result for classifying the Elliptical PNe. This morphological type was challenging. The accumulative average F1 score was the lowest among all of the PNe morphologies. Among the three DTL models, DenseNet201 was superior in classifying the Elliptical PNe with the average F1 score of 38%. MobileNetV2 gave an average F1 score of 36% and InceptionResNetV2 of 27%.

For InceptionResNetV2, the HASH DB images were not an effective image resource for classifying Elliptical PNe. The Pan-STARRS resources gave better results, with the Mask images being the most consistent between the three models. The other Pan-STARRS resources gave mixed results. InceptionResNetV2 wrongly classified most of the Pan-STARRS No-star Elliptical PNe images as Bipolar PNe. MobileNetV2 did a bit better here, but failed on the Pan-STARRS Quotient resource where most Elliptical PNe images were classified as Bipolar PNe.

In classifying the Round PNe (Figure 13), DenseNet201 DTL model was the best, with the average F1 score of 43% and a reasonably consistent performance. The second best model was MobileNetV2, with the average F1 score of 37% and lastly InceptionResNetV2 with the average F1 score of only 29%. All three models gave the best result for the HASH Optical images.

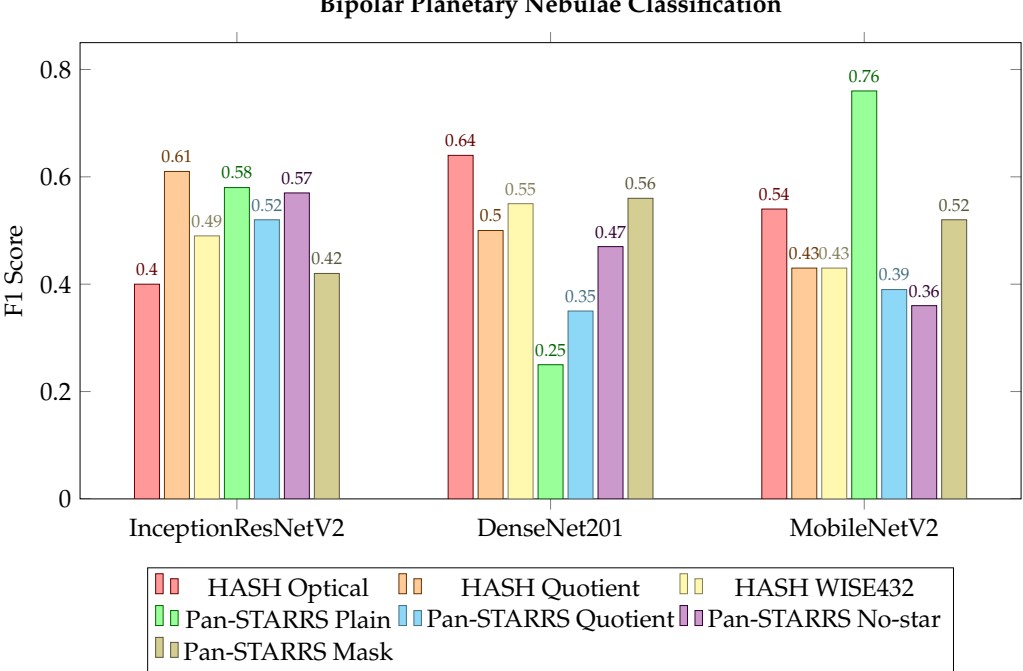

**Figure 11.** Bipolar planetary nebulae morphology classification F1 score using the Test set from HASH DB and Pan-STARRS.

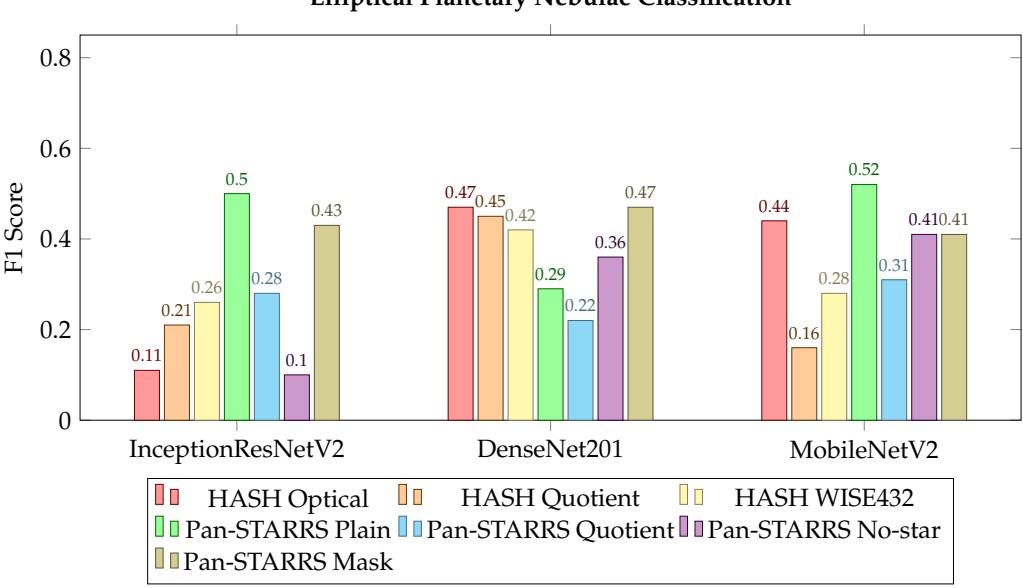

**Figure 12.** Elliptical planetary nebulae morphology classification F1 score using the Test set from HASH DB and Pan-STARRS.

Pan-STARR Mask was particularly poor for this morphological class, perhaps because the masks were themselves round. For this resource, both InceptionResNetV2 and MobileNetV2 classified most of the Round PNe images as Elliptical PNe. The DenseNet201 DTL model did best for this resource. The InceptionResNetV2 model had a problem with the WISE432 images. Visual inspection of the classification shows that none of the HASH WISE432 Round PNe images were correctly classified. Most of the classification fell into Bipolar PNe. Here, it should be noted that the dust emission measured by WISE may have a different morphological distribution than the gas emission measured by the other resources.

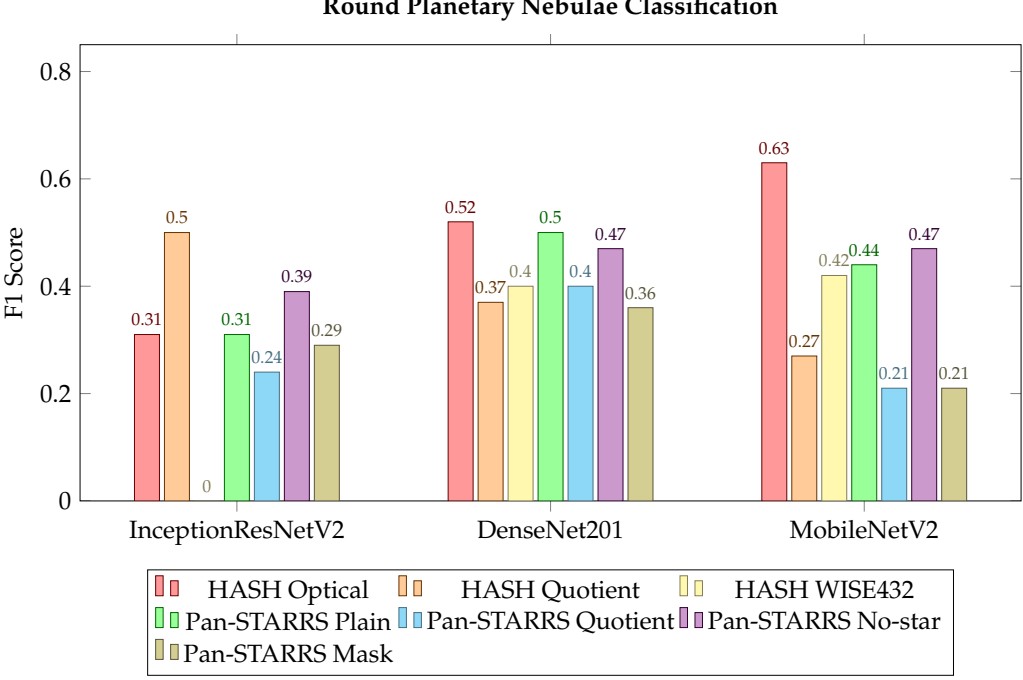

**Figure 13.** Round planetary nebulae morphology classification F1 score using the Test set from HASH DB and Pan-STARRS.

*3.5. Prediction of Morphologies*

Figure 14 shows an example confusion matrix, for the particular case of the DenseNet201 model and the HASH Optical images. This was chosen for having the highest F1 Score in Table 8 and Figures 11–13.

The confusion matrix indicates a reasonable result, with about half of objects correctly classified. The misclassified objects do not show a strong bias. The best identified category is that of the Bipolar nebulae, where two thirds are correctly classified. For Round and Elliptical nebulae about half are correct, with most of the confusion between the two categories. The conclusion is that the DTL has a good success on separating Bipolar PNe from the other two categories, but it is less successful distinguishing Round from Elliptical nebulae. If we combine Round and Elliptical into one group, then the Matthews correlation coefficient becomes $\phi = 0.45$.

**Predicted Morphology**

|  | Bipolar | Elliptical | Round |
|---|---|---|---|
| **Bipolar** | 19 | 7 | 5 |
| **Elliptical** | 5 | 13 | 9 |
| **Round** | 4 | 8 | 14 |

**Actual Morphology**

**Figure 14.** The confusion matrix of PNe morphology using HASH Optical with DenseNet201 DTL model predictions.

More accurate morphological classification may require higher quality image resources, especially to better separate Round from Elliptical PNe. Additional training of the DL model may also improve results. However, the current data show that the DTL models are able to do morphological classification, using the Optical images.

## 4. Discussion

Only a few studies have attempted ML aimed at PNe [8,9]. PN classification is a difficult problem, as PNe have a large variety in their appearances, can easily be confused with other types of objects (HII regions and galaxies, for instance) and are often faint objects located in dense star fields. We also did not use the highest quality data available, but used general-purpose surveys not optimized for PNe. In addition, we used transfer learning and did not fine-tune the parameters. The high success rate, with Matthews correlation coefficient of 90%, was therefore not necessarily expected. Classifying the morphologies was a more difficult task, but the results are promising.

Of the architectures that were tested, DenseNet201 was found to be the most consistent performer. InceptionResNetV2 also worked well, in some cases better than DenseNet201 but with variable results and at high computational cost, while MobileNetV2 was also acceptable but fell short in some tests on the morphology. Five other architectures were also tested (AlexNet, VGG-16, VGG-19, ResNet50 and NASNetMobile) but were found not to be optimal for this particular problem. Previous studies of ImageNet by Keras [23] have indicated that InceptionResNetV2, DenseNet201 and MobileNetV2 were among the best DL algorithms for image classification. We found that this also holds for astronomical images. For the implementation, we found that the Keras routine `predict()` produced significantly discrepant results from `evaluate()`, for unclear reason, and it could not be used. The MATLAB implementation produced consistent results.

The architectures were originally trained with the large ImageNet dataset and we did not optimize the parameters for our images. This transfer of learning is a limitation: it is plausible that results

will improve with a future training step which optimizes the feature extraction. Each algorithm also requires images of a specific maximum size, which is much smaller than typical astronomical images. This required some loss of resolution in some cases. Even with these drawbacks, we found strong results for the current sample. Combining four different diagnostic image resources, the Matthews correlation coefficient is an impressive 90%.

As a check, we inspected the images of objects that are cataloged as Rejected PN, but for which all four resources returned a classification as True PN. The objects are listed in Table 9. Of the eight objects in the table, in five cases, the available images do not suggest the target to be a PN. The fields are crowded, with multiple stars, infrared sources and in a few cases some extended emission, but the different tracers do not appear to centre on the same source. In three cases, the objects could be PNe, in two cases also with an indication from a spectrum. The third target shows an extended nebula. These three targets are worth further investigations.

We also used the DL algorithm to classify the samples of Possible PNe and Likely PNe. About half were classified as True PN. The ratio was twice as high among Likely PNe as opposed to Possible PNe. This agrees with expectations, as the level of confidence is higher for Likely PNe. This result is a good indication that the DTL is producing reasonable results.

**Table 9.** Rejected objects classified here as True PN. In the last column, 'p' indicates a potential PN while 'n' suggests a negative classification.

| PNG Number | Name | Visual Inspection |
| --- | --- | --- |
| 359.0+02.8 | Al 2-G | p |
| 001.0−02.6 | Sa 3-104 | p |
| 002.5−02.6 | MPA 1802−2803 | n |
| 001.8−05.3 | PM 1-216 | n |
| 002.4+01.4 | [DSH2001] 520-9 | n |
| 018.6-02.7 | PN PM 1-243 | n |
| 003.0−02.8 | PHR J1803−2748 | p |
| 140.0+01.7 | IPHASX J031434.2+594856 | n |

The second part of the this work focused on morphological classification of the True PNe. The results on this are best illustrated using Figure 14, albeit for only one resource. The correct classification was found in half of cases (for three possible classifications). The success rate was similar for Bipolar, Elliptical and Round. However, it was more difficult to distinguish Round from Elliptical nebulae. Although this is a reasonable result, morphological classification would benefit from better images than available from the surveys we used.

For future research, there are several aspects that can improve on the current result. The sample for non-PNe could have been improved, with clearly identified object types. This would allow classifying Rejected PNe into separate groups of objects, rather than a mixed bag of 'rejects'. A feedback step to optimize the learning to the specific images is also likely to improve the success rate. This includes K-fold Cross Validation and fine-tuning of the related hyper-parameters and model-parameters. Finally, a method that combines the diagnostics into a single training set, rather than analyzing them separately, may give even better results. This is also closer to how PNe are normally classified [36,37].

The research has shown that DL can identify and classify PNe. This first investigation is very promising and provides clear pathways for future research. PNe are among the most difficult problem for automated classification. This is therefore an important step in the application of DL in complex, wide-field astronomical images.

**Author Contributions:** Conceptualization, D.N.F.A.I. and A.A.Z.; methodology, D.N.F.A.I. and A.A.Z., R.A. and G.A.F.; software, D.N.F.A.I., I.M., A.H.F. and J.A.; validation, D.N.F.A.I., I.M. and A.A.Z.; formal analysis, D.N.F.A.I., A.A.Z. and I.M.; investigation, D.N.F.A.I., A.A.Z. and I.M.; resources, D.N.F.A.I. and A.A.Z.; data curation, D.N.F.A.I. and I.M.; writing—original draft preparation, D.N.F.A.I., A.A.Z. and I.M.; writing—review and editing, D.N.F.A.I., A.A.Z., I.M., R.A., G.A.F., A.H.F. and J.A.; visualization, D.N.F.A.I., A.A.Z. and I.M.;

supervision, A.A.Z.; project administration, A.A.Z. and D.N.F.A.I.; and funding acquisition, A.A.Z., D.N.F.A.I., R.A. and G.A.F. All authors have read and agreed to the published version of the manuscript.

**Funding:** This research was funded under the Newton program for the project entitled "Deep Learning for Classification of Astronomical Archives" under grant UK Science and Technology Facilities Council: ST/R006768/1 and the Newton-Ungku Omar Fund: F08/STFC/1792/2018.

**Acknowledgments:** This research would not have been possible without the exceptional support from the Ministry of Higher Education Malaysia, UK Science and Technology Facilities Council, Universiti Malaysia Sarawak, Universiti Sains Malaysia and the University of Manchester. This research has made use of the HASH PN database at hashpn.space and the Pan-STARRS1 Surveys (PS1). The PS1 surveys and the PS1 public science archive have been made possible through contributions by the Institute for Astronomy, the University of Hawaii, the Pan-STARRS Project Office, the Max-Planck Society and its participating institutes (the Max Planck Institute for Astronomy, Heidelberg and the Max Planck Institute for Extraterrestrial Physics), Garching, The Johns Hopkins University, Durham University, the University of Edinburgh, the Queen's University Belfast, the Harvard-Smithsonian Center for Astrophysics, the Las Cumbres Observatory Global Telescope Network Incorporated, the National Central University of Taiwan, the Space Telescope Science Institute, the National Aeronautics and Space Administration under Grant No. NNX08AR22G issued through the Planetary Science Division of the NASA Science Mission Directorate, the National Science Foundation Grant No. AST-1238877, the University of Maryland, Eotvos Lorand University (ELTE), the Los Alamos National Laboratory and the Gordon and Betty Moore Foundation.

**Conflicts of Interest:** The authors declare no conflict of interest. The funders had no role in the design of the study; in the collection, analyses, or interpretation of data; in the writing of the manuscript, or in the decision to publish the results.

## Appendix A. HASH DB Query

**Table A1.** Query submitted to obtain True PNe and Rejected PNe alongside other objects from HASH DB.

| Select Sample Options | True PNe | Rejected PNe and Other Objects |
|---|---|---|
| Status | True PN | Check all except True PN, Likely PN, Possible PN and New Candidates |
| Morphology | Check all | Uncheck all |
| Galaxy | Galactic PNe | Check all except Galactic PNe |
| Catalogs | Uncheck all | Uncheck all |
| Origin | Uncheck all | Uncheck all |
| Spectra | Uncheck all | Uncheck all |
| Checks | Uncheck all | Uncheck all |
| User Samples | Uncheck all | Uncheck all |

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
