# Peer review of "Classification of Planetary Nebulae through Deep Transfer Learning"

_galaxies, doi:10.3390/galaxies8040088_

Round 1

Reviewer 1 Report

This is an interesting paper that deals with the problem of automated classification of planetary nebulae (PNe) using Deep Learning (DL) algorithm. From the point of view of research on the field of PNe, I believe this work makes a progress in the field, although I have some concerns that I would like the authors to clarify and revise. 

I am particularlyconcerned about the image processing previous to the DL classification. The input consists of PNG images, with a 8-bit normalized intensity scale (255 levels). The authors will agree that a lot of information is lost with respect to the original FITS files. I consider this to be a serious drawback of the method here presented. More critical seems the complex image processing to remove background stars. This makes not universal the applicability of the method. It would be more useful to set limits to apply this method based on the stellar density (with respect to the nebular angular size).

The authors conclude that the DL method does not work properly for the morphological classification of PNe. In their words "Morphologies are probably best measured from targeted observations and it may be helpful to add such images to the training set." The first half of this statement does not come as a big surprise given the morphological and structural complexity of PNe. The need for additional information (deeper images, images in different emission lines) and the reduced number of PNe requiring morphological classification makes quite questionable even considering applying DL methods. It seems to me it is not worthwhile the amount of work devoted to this aim given the expected and obtained results.

Author Response

Please refer to the attached PDF file.

Reviewer 2 Report

The authors present a comparison of the performance of different combinations of DTL methods and set of images to identify and classify Planetary Nebulae. The manuscript is well written and mostly clear.

I have one major concern regarding the paper: I was left with the feeling that the authors are too optimistic on the final conclusion (first sentence in l. 559). I think they have proved that they can apply DTL methods to a set of images, and that the methods may be able to identify the PNe in the images (to some extent), but I do not think they have proven (yet) that these methods are able classify PNe (as they write in their l. 559). Well, of course, they provide a class for a given PNe, but I do not think that it can be trusted, at this stage. Given the many caveats and suggestions for improvements that the authors list, I suspect that they think similarly, only that for some reason they have not reflected that thought in the final conclusion. I think they should rephrase this so it reflects better the results that they present in the manuscript. I do not see anything wrong with concluding that the experiment performed here does not allow to say whether DTL methods can classify PNe. This is one of the first works where this kind of exercise is done, and there is added value by "only" presenting the exercise, saying that it is able to detect PNe (to some extent), and that the authors are not able to classify (yet) PNe with DTL methods. In that sense, the abstract is written in a more realistic manner, I think. So as long as they rephrase the last sentence following this spirit and take into account some minor comments that I will put below, I do consider the manuscript suitable for Galaxies. I will list here the minor comments in order of appearance:

l. 74. What is x_y_i? Shouldn't it be x_s_i? Maybe I missed something...

l. 107-108. When said "both IPHAS/VPHAS and SSS are available, we used the former". Why? Add a sentence with the motivation for this choice.

l. 120-121. It is fine to divide the images in Training 80%, Validation 10%, and Test 10%. It is quite standard. But I do not think one has to mention the Pareto principle for that. It refers to a different concept:
https://en.wikipedia.org/wiki/Pareto_principle

l. 130. This is the first time that the hyper-parameters are mentioned. I put this comment here, but not sure whether this is the best place to implement it. Maybe it is better in Sec 2.3, when the models are described, and perhaps a bit in the discussion / conclusions too. Here, it is said that the hyper-parameters are not tuned. And in the conclusions / discussion, it is mentioned that one possibility for improvement would be the tuning of these hyper-parameters. I miss a description of what are the hyper-parameters available for each of the three selected DL algorithms (at least). I see that there is something about this around l. 306, but it is difficult to infer the meaning of the hyper-parameters there. Things like RMSprop(lr=1e-4) are not with obvious meaning. Also, it is not clear which are the most critical ones and whether they are the only hyper-parameters that can be tuned.

In that sense, it is my understanding that not all the hyper-parameters will have the same impact in the performance of the DL algorithm. It is mentioned that the authors have not done a thorough tuning of these hyper-parameters, but if for some reason they have done any preliminary test that suggest promising perspectives for the tuning of this or that hyper-parameter, I would suggest to include a few sentences about that in the manuscript.

l. 215 Specify here which category 'Possible' or 'Likely' has higher degree of confidence. It is said later on (page 15), but here is the first appearance of the categories, I think.

l. 260 Even if the five rejected algorithms are not listed in table 5, I think it would be useful to have an idea of their Top-1 Accuracy, for reference and comparison. I would provide range or typical value here.

Sec 3. General. When discussing the results (Table 7, Fig 7, Table 8, Figs 11, 12, 13), sometimes the difference between the best option and the second best option is minimal (e.g. 0.81 vs. 0.82). The authors mention something about that, I guess in l. 331. But I do not see anywhere a discussion or a mention to the number that makes two results statistically different. I mean: in the previous example, the authors would write that the model with 0.82 (according to a given metric) is better than the model with 0.81. But this is true if two models with differences e.g. >0.009 are statistically significant. For statistically significant differences e.g. >0.050, these two models would be equally good, and both should be put in bold / underlined in the corresponding table (if they are the largest numbers). So... what is the number that makes the metric of two given models different?

l. 354. The authors have decided to combine the four predictions with equal weights to produces the final class. However, if I understood correctly, previous results indicate that not all the set of images are equally good. (From ~better to ~worse: HASH Optical->HASH Quotient->HASH WISE-> PAN-STARRS). Why did the authors decide to apply equal weights and not some weights based on these results? Please clarify. Also, with these weights the situation of having a PN with equal classification probability would disappear (or at least would be less likely).

l. 369. I think "relative disappointing" is too optimistic for this result. This is too close to random!

l. 382. I would replace PN for "target" or "candidate" or "object", since the authors are classifying it.

l. 386. I suspect that the fact that half of them could not be classified could change/improve if the weights are changed as suggested for comment in l. 354...

l. 399. I agree that having 100% accuracy, precision and recall sounds to much to overfitting... Only for this particular case... Have the authors tried a model with, say, 50 epochs only, to see if training and test sample give more consistent results?

l. 465. It is written "... having the highest F1 (accuracy) score" as if F1 score and accuracy were the same concept. But equations 4 and 7 show that they are two different things. What do the authors mean here? Clarify.

l. 479. The authors state in their conclusions that to their knowledge this is the first ML study on PNe. To my knowledge there are not many, but at least I know this other one (1996 !!!):
https://ui.adsabs.harvard.edu/abs/1996A%26AS..116..395F/abstract
Also, this one, more recent:
https://ui.adsabs.harvard.edu/abs/2019MNRAS.488.3238A/abstract

l. 572. The webpages for HASH PN database and Pan-STARRS mention what should be included in the acknowledgements. I copy and paste as it is there, so the authors fix this section accordingly but I think the part for the references is already implemented.

* HASH PN database
If you use this resource in a publication, please cite this paper:Parker, Bojičić & Frew 2016 <http://adsabs.harvard.edu/abs/2016arXiv160307042P>and include the following acknowledgement: "This research has made use of the HASH PN database at hashpn.space”.

*Pan-STARRS1 Surveys (PS1)
Credit where it is due

Here is the text for acknowledging PS1 in your publications:

The Pan-STARRS1 Surveys (PS1) and the PS1 public science archive have been made possible through contributions by the Institute for Astronomy, the University of Hawaii, the Pan-STARRS Project Office, the Max-Planck Society and its participating institutes, the Max Planck Institute for Astronomy, Heidelberg and the Max Planck Institute for Extraterrestrial Physics, Garching, The Johns Hopkins University, Durham University, the University of Edinburgh, the Queen's University Belfast, the Harvard-Smithsonian Center for Astrophysics, the Las Cumbres Observatory Global Telescope Network Incorporated, the National Central University of Taiwan, the Space Telescope Science Institute, the National Aeronautics and Space Administration under Grant No. NNX08AR22G issued through the Planetary Science Division of the NASA Science Mission Directorate, the National Science Foundation Grant No. AST-1238877, the University of Maryland, Eotvos Lorand University (ELTE), the Los Alamos National Laboratory, and the Gordon and Betty Moore Foundation.

In addition, please cite the following papers describing the instrument, survey, and data analysis as appropriate:

1. The Pan-STARRS1 Surveys <https://arxiv.org/abs/1612.05560>,
Chambers, K.C., et al.
2. Pan-STARRS Data Processing System
<https://arxiv.org/abs/1612.05240>, Magnier, E. A., et al.
3. Pan-STARRS Pixel Processing: Detrending, Warping, Stacking
<https://arxiv.org/abs/1612.05245>, Waters, C. Z., et al.
4. Pan-STARRS Pixel Analysis: Source Detection and Characterization
<https://arxiv.org/abs/1612.05244>, Magnier, E. A., et al.
5. Pan-STARRS Photometric and Astrometric Calibration
<https://arxiv.org/abs/1612.05242>, Magnier, E. A., et al.
6. The Pan-STARRS1 Database and Data Products
<https://arxiv.org/abs/1612.05243>, Flewelling, H. A., et al.

Author Response

Please refer to the attached PDF file.
